# SIGNALS OF TRUST: FROM CLASSIFICATION TO CONFIDENCE

## ABSTRACT

Reliable confidence estimation is crucial for safety-critical and human-in-the-loop applications, where users must trust individual AI decisions under uncertainty. Existing methods, ranging from softmax-based probabilities to uncertainty quantification techniques, fall short in providing reliable confidence scores. These scores often suffer from poor calibration, high computational costs, or lack of interpretability, making them less effective at the instance level. To overcome these limitations, we introduce **KDE Trust**, a post-hoc method that models class-conditional distributions of correct and incorrect predictions in the deep neural network's representation space. This yields a direct estimate of the posterior probability of correctness, $P(\text{correct}|\text{predicted} = c)$, without requiring architectural modifications. We also introduce a novel evaluation metric called Polarisation, which quantifies how close confidences scores are to 1 for correct classifications and 0 for incorrect ones, enabling human-centric interpretability. Across four datasets spanning high- and low-accuracy regimes, as real-word scenarios often involve suboptimal model performance, KDE Trust matches baseline performance in high-accuracy settings while achieving up to +14.2% AUROC improvement in degraded scenarios. Precision–percentile curves show higher retained precision with reduced variance, and distributional metrics reveal polarised scores.

## 1 INTRODUCTION

The rapid integration of Artificial Intelligence (AI) systems into critical areas such as medical diagnosis, industries, autonomous vehicles, and defence (The White House, 2025; European Defence Agency, 2025) demands reliability that extends beyond overall accuracy, because users must be able to trust the system's decisions even under uncertain or challenging conditions (Li et al., 2021). A major challenge is that deep learning models always output a prediction, even in situations of high uncertainty or when they lack sufficient information, rather than admitting uncertainty or abstaining from a response (Schreck et al., 2024). To establish appropriate trust and foster effective human-AI collaboration, it is imperative that human operators be able to assess the reliability of each individual AI decision (Afroogh et al., 2024). However, existing approaches struggle to provide an interpretable and per-instance confidence score, leaving a gap in the toolkit for trustworthy AI. This work proposes a new approach to fill this gap.

This framework consists of estimating confidence for individual classifications, i.e., the probability that a given prediction is correct, conditioned on the class assigned by the model. Our approach explicitly models the distribution of correctly classified and misclassified examples for each predicted class, providing a score that is directly interpretable for the user. This idea is actually inspired by the human experience, where confidence in actions is built upon prior experiences and successful outcomes. Just as humans become more confident in their decisions when they have faced similar situations with positive results, we aim to apply a similar philosophy to machine learning, specifically to deep neural networks (DNNs) and their classification tasks. By leveraging this intuition, our framework seeks to provide reliable confidence scores for AI classifications.

To achieve this, our method distinguishes between reliable and unreliable classifications by modelling two distinct distributions for each class based on the training data: one representing the patterns of correctly classified examples and another capturing the patterns of misclassified examples. When the model makes a new prediction, our framework compares the example's internal represen-

tation (e.g., logits or deep features) to these two distributions. The relative similarity between the new representation and the correct/incorrect distributions directly reflects the model's confidence in the classification's correctness. Specifically, the closer the representation aligns with the correct distribution (compared to the incorrect one), the higher the confidence score. This comparison leverages the training data's inherent statistics to estimate $P(\text{correct}|\text{predicted} = c)$.

## 2 RELATED WORK

Confidence estimation is a long-standing challenge in machine learning, with various approaches emerging to address this problem. This section provides an overview of the related work, highlighting the strengths and limitations of existing methods and positioning our KDE Trust in the context of these contributions.

**Confidence Estimation in Classical Machine Learning**   Traditional machine learning models provide well-established methods for quantifying confidence. Support Vector Machines (SVMs) produce decision scores that can be interpreted as measures of confidence, though they lack probabilistic semantics without post-processing (Platt et al., 1999). Random Forests yield vote-based probability estimates, which are intuitive but often require refinement to achieve reliable calibration (Breiman, 2001). Bayesian models directly provide posterior probabilities under explicit modelling assumptions, offering interpretable confidence when the model's specifications hold (Pearl, 2014). While these methods enable confidence quantification, they share a fundamental limitation: the scores they produce do not inherently align with the true likelihood of correctness. This misalignment between predicted confidence and actual reliability motivated the calibration techniques.

**The Rise of Calibration Techniques**   Calibration methods emerged to address this limitation by adjusting model outputs to ensure that predicted probabilities reflect empirical frequencies (Niculescu-Mizil & Caruana, 2005). Techniques such as Platt scaling (Platt et al., 1999), isotonic regression (Zadrozny & Elkan, 2001), and temperature scaling (Guo et al., 2017) became standard tools for improving the reliability of confidence estimates. However, calibration remains an aggregate property: it corrects global misalignment but may obscure subpopulation-level miscalibration or fail to capture instance-specific nuances. These limitations are particularly problematic for modern DNNs, where softmax-based confidence scores are notoriously miscalibrated and overconfident (Guo et al., 2017).

**Challenges in Deep Learning and Uncertainty Quantification**   The calibration challenges inherent to deep learning have spurred the development of dedicated uncertainty quantification (UQ) methods. Approaches such as Monte Carlo Dropout (Gal & Ghahramani, 2016), Deep Ensembles (Lakshminarayanan et al., 2017), and Bayesian neural networks attempt to address both aleatoric and epistemic uncertainty. While these methods provide more nuanced uncertainty estimates, they often incur significant computational overhead and do not always yield directly interpretable probabilities for individual predictions (Abdar et al., 2021). Moreover, they typically require architectural modifications or multiple forward passes, limiting their practicality in resource-constrained settings.

**Explanations, conformal prediction and instance-level confidence**   Explainable AI (XAI) techniques provide insights into model decision-making, but do not certify per-instance correctness and can inadvertently increase user overconfidence if explanations are plausible despite incorrect predictions (Ribeiro et al., 2016; Spitzer et al., 2025).

Conformal prediction provides model-agnostic prediction sets with guaranteed coverage for a chosen confidence level, offering formal guarantees but answering "which labels are plausible?" rather than giving a probability that the top label is correct (Shafer & Vovk, 2008; Angelopoulos & Bates, 2022).

Other recent methods attempt to assign confidence scores to individual predictions:

- **Trust Score** (Jiang et al., 2018): computes a consistency-based measure by comparing distances in feature space between the test example and high-density regions of the predicted class vs other classes. While the Trust Score captures geometric conformity, its unbounded

nature limits interpretation and prevents direct use as a normalised confidence score between in $[0, 1]$.

- **MAPLE** (Venkataramanan et al., 2023): regularises the latent space (via triplet loss) so that representations of each class become approximately Gaussian. At inference, confidence is derived from the Mahalanobis distance between the instance and class centroids. However, MAPLE requires additional training overhead to impose these latent space properties.

**Gap and positioning**   In short, classical models and calibration provide established tools for probabilistic outputs, and modern UQ methods offer rich uncertainty estimates for deep models. However, no prevailing approach supplies a single, normalised, instance-level confidence score for the predicted label in deep neural classifiers such as CNNs or BERT. Existing methods either provide global guarantees (CP), explain decisions without quantifying correctness (XAI), or assign geometric conformity scores that are unbounded or require costly re-training (e.g., Trust Score, MAPLE).

Also, confidence estimation techniques are often tested on high-accuracy models, but this doesn't reflect many real-world scenarios. In reality, achieving high accuracy can be difficult due to limited data, computational resources, or task complexity. Reliable confidence estimation is still crucial in these cases for safe decision-making. Our method addresses this need by introducing a post-training confidence score that is easy to interpret and works with any deep classifier, regardless of its accuracy: from state-of-the-art performance to reasonable lower accuracy levels (e.g., up to 25-30 percentage points (pp) below peak performance).

To further support human interpretability, a critical but underexplored aspect in confidence estimation, we introduce a novel evaluation metric called Polarisation. This metric quantifies how close confidence scores are to 1 for correct classifications and 0 for incorrect ones. Polarisation offers a practical tool to assess the human usability of confidence estimation methods.

While much work evaluates confidence via out-of-distribution (OOD) detection (Yang et al., 2024; Berger et al., 2021), our focus is distinct: we assess a model's ability to rank in-distribution (ID) predictions by correctness. A critical need for the applications like those mentioned earlier, where users must trust individual decisions on typical inputs. We prioritise ID evaluation because OOD detection and confidence ranking address different goals (anomaly detection vs. reliability assessment). Though we do not evaluate OOD performance here, our approach is compatible with existing OOD techniques and could be extended in future work.

## 3 MODELLING

### 3.1 THE FUNDAMENTAL IDEA: LOCALISATION OF CHARACTERISTICS IN SPACE

Our approach to quantifying confidence is based on a simple but powerful intuition: if the representation of an instance for a predicted class $c$ is located in a region with a high density of correctly classified examples for that class, this indicates high confidence. Conversely, if it lies in a region with a high density of misclassified examples, this suggests low confidence. The goal is therefore to model these underlying distributions to evaluate the typicality of a new example relative to the model's past behaviour. To do this, we construct two density estimators for each predicted class $c$:

$$\widehat{f}_{\text{good},c}(x) : \text{the estimated density on the correctly classified examples of class } c,$$

$$\widehat{f}_{\text{bad},c}(x) : \text{the estimated density on the incorrectly classified examples of class } c.$$

Several density estimation methods could be used to model these distributions. For example, Gaussian distributions is an option, but they require specialised training objectives like triplet loss (Venkataramanan et al., 2023). However, such requirements are incompatible with our post-hoc setting. Similarly, Gaussian Mixture Models (GMMs), which approximate complex distributions using multiple Gaussian components, pose implementation challenges: they can introduce irreducible bias if poorly specified, even with large sample sizes (Dwivedi et al., 2018). Kernel Density Estimation (KDE), by contrast, is a non-parametric solution that makes no assumptions about the underlying data distribution. This allows KDE to converge to any density shape with a sufficient number of samples, without requiring prior knowledge of the distribution's form or the number of components (Chen, 2017). We therefore adopt KDE as our density estimation method for the remainder of this work. For the theoretical aspects of KDE and their application in our case, see Appendix A.

## 3.2 BAYESIAN FRAMING

Let $c$ denote the model's predicted class for an input $x$, where $x$ corresponds to any layer of the representation space of the DNN. We are interested in the posterior probability that the prediction is correct, conditioned on the predicted class $c$:

$$P(C_{\text{good}} \mid x, c)$$

where $C_{\text{good}}$ (resp. $C_{\text{bad}}$) denotes the event that the prediction for class $c$ is correct (resp. incorrect).

By Bayes' rule, this probability can be expressed as:

$$P(C_{\text{good}} \mid x, c) = \frac{f_{\text{good},c}(x) \, P(C_{\text{good}} \mid c)}{f_{\text{good},c}(x) \, P(C_{\text{good}} \mid c) + f_{\text{bad},c}(x) \, P(C_{\text{bad}} \mid c)} \tag{1}$$

where:

- $f_{\text{good},c}(x)$ and $f_{\text{bad},c}(x)$ are the class-conditional densities of $x$ for correct and incorrect predictions,
- $P(C_{\text{good}} \mid c)$ is the prior probability that a prediction for class $c$ is correct (i.e., the model's class-conditional accuracy for class $c$),
- $P(C_{\text{bad}} \mid c) = 1 - P(C_{\text{good}} \mid c)$.

This formulation highlights that the posterior probability is a function of the ratio of the densities $f_{\text{good},c}(x)$ and $f_{\text{bad},c}(x)$, weighted by the prior odds $\frac{P(C_{\text{good}}|c)}{P(C_{\text{bad}}|c)}$.

## 3.3 KDE TRUST: CONSTRUCTION AND IMPLEMENTATION

We do not assume analytic forms for $f_{\text{good},c}$ and $f_{\text{bad},c}$. Instead we estimate them from training data (restricted to examples whose predicted class is $c$) using class-conditional KDEs. Concretely, for each predicted class $c$, we fit two KDEs on the model representation of training examples: one on correctly classified examples to obtain $\widehat{f}_{\text{good},c}$, and one on misclassified examples to obtain $\widehat{f}_{\text{bad},c}$.

By substituting the true densities and priors with their empirical estimates in the Bayesian formulation, we obtain the following practical scoring rule:

$$\text{score}_{\text{KDE}}(x) = \frac{\widehat{f}_{\text{good},c}(x) \, \widehat{P}(C_{\text{good}} \mid c)}{\widehat{f}_{\text{good},c}(x) \, \widehat{P}(C_{\text{good}} \mid c) + \widehat{f}_{\text{bad},c}(x) \, \widehat{P}(C_{\text{bad}} \mid c)} \tag{2}$$

where $\widehat{P}(C_{\text{good}} \mid c)$ and $\widehat{P}(C_{\text{bad}} \mid c)$ are empirical estimates of the prior probabilities (e.g., computed from training data counts), and $\widehat{P}(C_{\text{bad}} \mid c) = 1 - \widehat{P}(C_{\text{good}} \mid c)$.

This score represents the estimated posterior probability that the prediction for class $c$ is correct, given the input $x$.

**Equal-prior simplification.** While the Bayesian formulation naturally involves class-dependent priors, in practice, we often adopt the equal-prior simplification $\widehat{P}(C_{\text{good}} \mid c) = \widehat{P}(C_{\text{bad}} \mid c)$ (Lazarow et al., 2017; Kumar et al., 2024; Hofmann et al., 2024). This reduces the scoring rule to a ratio of estimated densities:

$$\text{score}_{\text{KDE}}(x) = \frac{\widehat{f}_{\text{good},c}(x)}{\widehat{f}_{\text{good},c}(x) + \widehat{f}_{\text{bad},c}(x)} \tag{3}$$

This choice offers several practical advantages: (i) it avoids the need to estimate priors, which can be unstable in low-data regimes or under domain shift, (ii) it yields a geometry-driven score that directly reflects the relative likelihood of a point under the good vs. bad densities, and (iii) it produces a confidence score that is independent of the model's overall accuracy, an important property for human-in-the-loop applications where trust calibration should not be conflated with raw performance.

We therefore adopt the equal-prior variant in our experiments. Nevertheless, in Appendix B we investigate the impact of this assumption.

### 3.4 Assessment Protocol

To evaluate the effectiveness of a confidence score, we employ three complementary metrics that collectively assess its ability to discriminate between correct and incorrect predictions, demonstrate distributional separation, and align with human expectations, each addressing a distinct aspect of confidence score quality.

**AUROC (Area Under the Receiver Operating Characteristic)**   This metric assesses how well confidence scores separate correct from incorrect predictions, independently of the decision threshold. Formally, AUROC corresponds to the probability that a randomly chosen correct prediction receives a higher confidence score than a randomly chosen incorrect one (Fawcett, 2006). As shown in Hendrycks & Gimpel (2016), AUROC is a standard, threshold-free metric for evaluating detectors of misclassified examples, making it directly suited to our setting. See Appendix C.1 for formal definition.

**Wasserstein distance (or Earth Mover's Distance)**   Unlike traditional distance measures, Wasserstein distance measures the minimal "cost" of transforming one probability distribution into another, making it ideal for quantifying the separation between confidence scores for correct and incorrect predictions: a higher value signals better discrimination (Panaretos & Zemel, 2019). This metric is already proven efficient in CP and OOD detection (Xu et al., 2025; Wang et al., 2023). See Appendix C.2 for formal definition.

**Polarisation**   A metric we introduce to quantify the interpretability of confidence scores. The Polarisation score measures how closely correct predictions cluster around 1 and incorrect predictions around 0. A high score indicates indicates stronger Polarisation. Formally, let $c_i \in [0, 1]$ denote the confidence score of prediction $i$, and $y_i \in \{0, 1\}$ its correctness indicator ($y_i = 1$ if correct, 0 otherwise). Define the sets $C = \{i \mid y_i = 1\}$ and $I = \{i \mid y_i = 0\}$, with cardinalities $N_c = |C|$ and $N_i = |I|$. The Polarisation score is given by

$$\text{Polarisation} \; = \; \tfrac{1}{2}\left[ 1 - \frac{1}{N_c}\sum_{i \in C}|c_i - 1| \; + \; 1 - \frac{1}{N_i}\sum_{i \in I}|c_i - 0| \right] \tag{4}$$

which returns a value in $[0, 1]$.

## 4 Experiments

Our KDE Trust score approach is put to the test through an extensive experimental evaluation on image and text classification tasks, where we compare its ability to produce high-quality, interpretable confidence scores to that of the Softmax baseline and the Trust Score (Jiang et al., 2018), which represents a closely related state-of-the-art method. We do not include MAPLE (Venkataramanan et al., 2023) in our comparisons, as it is primarily designed for OOD detection and relies on additional representation learning with metric-based losses. Since our focus is on post-training, instance-level confidence estimation for ID predictions, MAPLE is less directly comparable to our setting. While confidence calibration is also a critical advancement for aligning predicted probabilities with empirical frequencies, it fundamentally differs from our goal of polarisation: explicitly separating correct and incorrect predictions to enhance instance-level interpretability. Also, since calibration preserves score rankings and thus does not affect ranking-dependent metrics (e.g., AUROC, precision in percentile level), we do not compare against it.

### 4.1 Datasets and Settings

Our evaluation covers four benchmark datasets (image and text classification) in Table 1, with model architectures and accuracy regimes in Table 2.

To induce low-accuracy regimes, we intentionally degraded model performance:

- Early stopped at 1 epoch.
- Used limited training batches (250 for MNIST and CIFAR-10, 25 for FashionMNIST).

Table 1: Summary of datasets used in our experiments.

| Dataset | Modality | Train / Test Size | #Classes |
|---|---|---|---|
| MNIST (LeCun et al., 2010) | Grayscale images (28×28) | 60k / 10k | 10 |
| FashionMNIST (Xiao et al., 2017) | Grayscale images (28×28) | 60k / 10k | 10 |
| CIFAR-10 (Krizhevsky, 2009) | RGB images (32×32) | 50k / 10k | 10 |
| IMDB (Maas et al., 2011) | Text reviews | 25k / 25k | 2 |

Table 2: Model architectures and accuracy regimes.

| Dataset | Architecture | High Accuracy | Low Accuracy |
|---|---|---|---|
| MNIST | 2-layer CNN | 98% | 76% |
| FashionMNIST | Resnet-18 (He et al., 2016) | 91% | 69% |
| CIFAR-10 | Resnet-18 (He et al., 2016) | 89% | 49% |
| IMDB | DistilBERT-base-uncased (Devlin et al., 2018) | 92% | 82% |

- Employed a high learning rate for IMDB (2e-3 instead of 2e-5).

This setup, spanning high-accuracy models (up to 98%) to severely degraded ones (down to 49%), enables a comprehensive evaluation of our method's robustness across accuracy regimes, a critical requirement since real-world applications often rely on models with varying performance levels, where confidence must remain reliable regardless of underlying accuracy.

## 4.2 CONFIGURATION AND ABLATIONS

Our KDE Trust score relies on three key components: the kernel $K$, bandwidth estimator $H$, and distance metric $D$. We performed a systematic ablation across all datasets, evaluating two kernels (Gaussian, Exponential), three bandwidth estimators (Silverman, Scott, Grid Search), and three distance metrics (Euclidean, Manhattan, Mahalanobis). Details are reported in Appendix D. The best-performing configuration combines a **Gaussian kernel**, **Silverman's rule** for bandwidth, and the **Mahalanobis distance metric**, which matched or outperformed alternatives across all evaluation metrics.

We also assessed different model layers as the feature representation ($x$) for KDE Trust : the logit layer, intermediate layers (penultimate $n-1$ and antepenultimate $n-2$), and the input representations (for BERT, the `[CLS]` token). Due to the high dimensionality of intermediate and input-layer features, we applied PCA with 10 components, which proved sufficient; increasing to 20 components yielded no further improvement. Appendix E presents visual comparisons across layers, where the **logit layer** emerged as the most effective and is used for all subsequent experiments.

## 4.3 RESULTS ON BENCHMARK DATASETS

To evaluate the performance of our KDE Trust score, we conducted experiments on the datasets summarised in Table 1, using both high-accuracy and low-accuracy model regimes (see Table 2). Our approach is benchmarked against the Softmax response and the Trust Score (Jiang et al., 2018) under both regimes. For the Trust Score, we adhered to the authors' recommendations, using the official implementation with parameters $k = 10$ and $\alpha = 0$. All reported results are averaged over five random seeds (0, 1, 2, 3, 42) to ensure robustness.

Given the Trust Score's unbounded range ($[0, +\infty)$) and outlier prevalence, we winsorised its trust values at the $5^{\text{th}}$–$95^{\text{th}}$ percentiles before scaling to enable fair comparisons with bounded metrics like Polarisation and Wasserstein distance. Figures 1–3 present a detailed comparison of KDE Trust against the baselines across both accuracy regimes. Appendix F provides detailed numerical results for all experiments.

**Results Analysis**  In high-accuracy settings, KDE Trust is on par with the strongest baseline in AU-ROC (Figure 1), with a minimal average difference of -0.6%. However, in low-accuracy regimes,

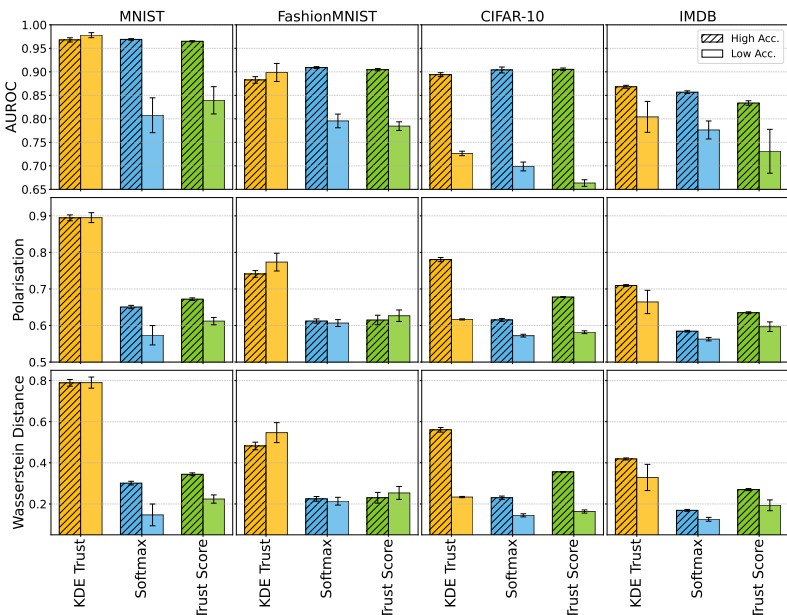

Figure 1: Comparison of the proposed **KDE Trust** score against Softmax and Trust Score baselines. **High-accuracy regimes** (striped bars), KDE Trust performs comparably to baselines in terms of AUROC, with a marginal average decrease of -0.6%, while achieving significantly higher separation, reflected in average gains of 16% in polarisation and 45% in Wasserstein distance. **Low-accuracy regimes** (solid bars) reveal significant improvements, with AUROC gains ranging from +3.5% to +14.2% over the highest-performing baseline. KDE Trust also achieves superior Polarisation (+17% avg.) and Wasserstein distance (+49% avg.) across all accuracy regimes. All values are reported with 95% confidence intervals.

our method yields substantial gains, increasing AUROC by +8.2% on average. Furthermore, precision–percentile curves (Figure 2) confirm this trend. KDE Trust maintains markedly higher precision in low-accuracy settings, crucial for reliability-dependent tasks where incorrect classifications carry severe consequences. In high-accuracy settings, the curves converge, indicating similar performance. Notably, our method also exhibits narrower variance bands (29% reduction compared to the next best-performing baseline in variance), indicating greater robustness. Beyond AUROC, separation metrics (Figures 1 and 3) highlight KDE Trust's stronger discriminative power. In particular, correct and incorrect predictions form clearly separated, polarised distributions even under low accuracy, whereas baselines collapse into overlapping, poorly polarised scores. Quantitatively, our method improves Polarisation by +16.5% on average and Wasserstein distance by +47.2%, across all accuracy regimes. However, we observe a significant drop in performance across all metrics when moving from high-accuracy to the highly degraded scenario (CIFAR-10, -40pp), underscoring the challenges of confidence estimation in this setting.

To further support interpretability, we also provide a visual illustration of how individual predictions are positioned in the density space from which the confidence score is derived in Figure 4.

## 5 CONCLUSION

We introduced **KDE Trust**, a simple and architecture-agnostic confidence score that models class-specific distributions of correct and incorrect predictions via KDEs. Without modifying the underlying model, it produces discriminative and interpretable scores grounded in the logits layer.

KDE Trust provides a measure of model reliability that aligns with the strongest baselines in AUROC in high-accuracy regimes (with <1% loss) and achieves gains of +3.5% to +14.2% in low-accuracy regimes. This consistency, in contrast to other methods, positions it as a general-purpose confidence metric suitable for a wide range of model accuracy levels. However, performance de-

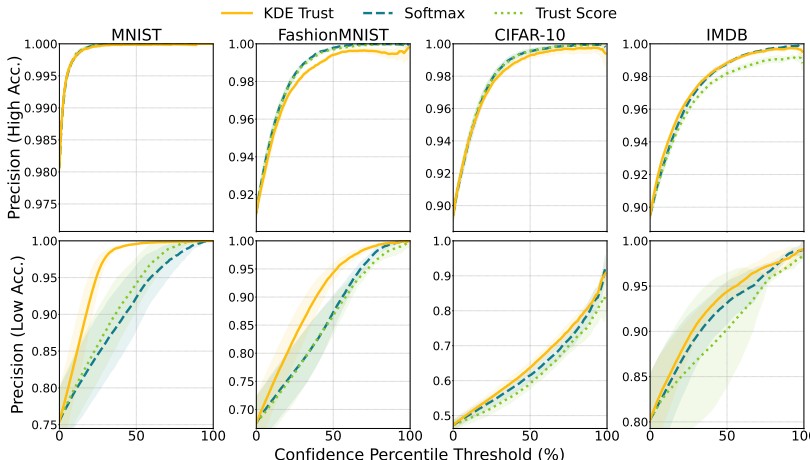

Figure 2: Precision as a function of the percentile confidence threshold for discarding the least con-fident predictions. **Axes**: The X-axis indicates the percentile threshold (0 = all predictions retained, 100 = only the most confident retained), and the Y-axis shows the precision of retained predictions (0–1). **High-accuracy regimes** (first row): KDE Trust performs on par with or marginally below baselines, with a maximum precision difference of -0.6% observed at the 39th percentile on Fashion-MNIST. **Low-accuracy regimes** (second row): KDE Trust consistently outperforms all alternatives, with notably smaller variance bands, demonstrating improved robustness.

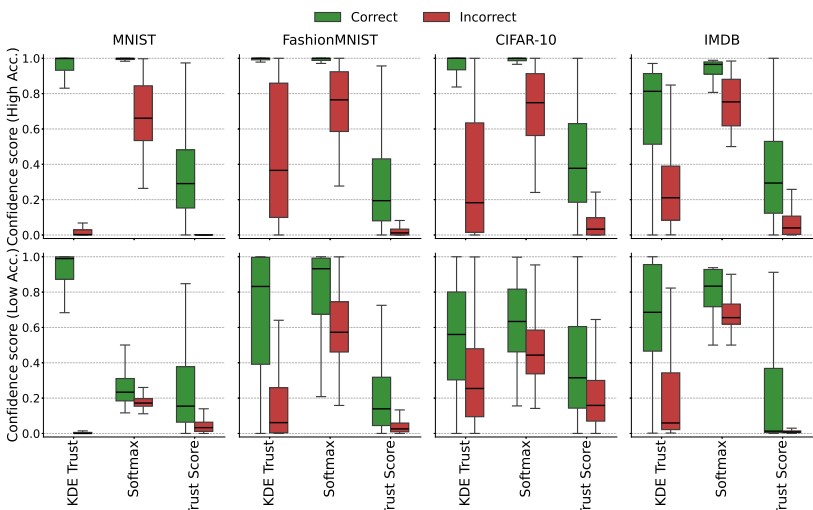

Figure 3: Boxplots of confidence scores for correct and incorrect predictions. The first row shows **high-accuracy regimes**, where KDE Trust achieves the strongest separation between correct and incorrect predictions. The second row shows **low-accuracy** regimes, which are more challenging (wider distributions, lower medians for correct predictions). While our method generally maintains polarised and well-separated scores in these regimes, CIFAR-10 with only 49% accuracy stands out as a demanding configuration, where KDE Trust still exhibits better Polarisation and Wasserstein distance than baselines despite overlapping boxplots.

clines in very low-accuracy regimes (<50% accuracy), where model uncertainty becomes extreme, highlighting the need for complementary approaches in such scenarios.

Precision–percentile curves show that KDE Trust retains higher precision with reduced variance across benchmarks compared to alternative methods. This consistency in precision is particularly valuable in domains where minimising false positives is critical, such as healthcare, autonomous systems, or safety-critical applications. Furthermore, our proposed Polarisation metric quantifies

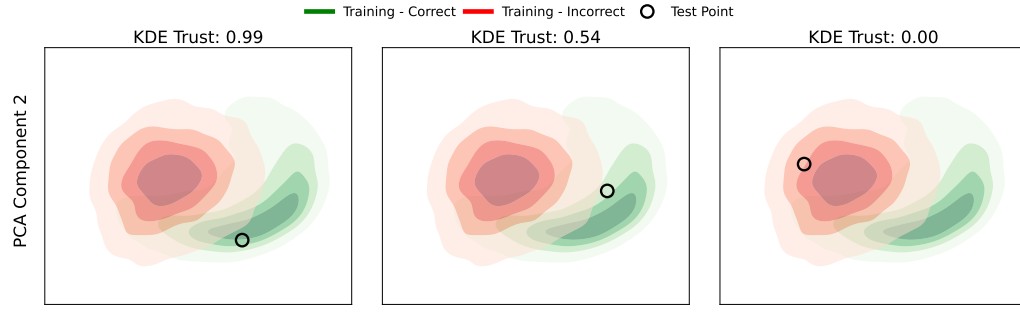

Figure 4: Illustration of the density-space interpretation of KDE Trust on dataset MNIST, seed 0 for three different predictions of class 1 by the model. **Left**: an example with a correct classification with a high confidence score, located in a high-density region of correct classifications. **Middle**: a borderline case with a score around 0.5, lying between the densities of correct and incorrect classifications, corresponding to a correctly classified prediction. **Right**: a low-confidence example positioned in a region dominated by incorrect classifications, corresponding to a misclassified prediction.

how effectively confidence scores distinguish between correct and incorrect classifications, with scores clustering near 1 for correct predictions and 0 for incorrect ones. This distributional clarity not only validates the relevance of KDE Trust's score distribution but also enhances its human interpretability, ensuring alignment with intuitive expectations of confidence.

Naturally, KDE Trust can be integrated with calibration and uncertainty quantification (UQ) methods to form a comprehensive reliability assessment framework. Its instance-level, human-readable scores make it compatible with XAI tools, enabling end-users to interpret confidence estimates alongside model explanations. For example, it could support prioritisation of predictions for human review, adaptation of intervention thresholds in dynamic systems, or enhancement of explanations with a measure of intrinsic reliability.

**Perspectives** The future directions are: (i) developing online adaptation mechanisms that take human feedback into account (incremental relabelling of KDEs), (ii) explicitly combining OOD detection with our score to enhance security, (iii) extending the method to more complex outputs (e.g. LLMs), (iv) integrating post-hoc calibration methods to enable strict probabilistic interpretation where required, and (v) exploring active learning scenarios, where our confidence measure can naturally guide the selection of informative samples for annotation.

REPRODUCIBILITY STATEMENT

We took several measures to ensure the reproducibility of our results. All datasets used (MNIST, FashionMNIST, CIFAR-10, and IMDB) are standard benchmarks, loaded directly via the Hugging Face dataset library and PyTorch (see Table 1). The proposed KDE Trust score is formally defined in Section 3.3, with complete mathematical details provided in Appendix A, and visual illustrations in Figure 4. All experiments, including both the main evaluations (cf. Appendix F) and the ablation study (cf. Appendix D), are averaged over five seeds to ensure robustness (seeds 0, 1, 2, 3, 42). Full implementation details, including model training, evaluation protocols, and plotting scripts, are available at `https://anonymous.4open.science/r/KDE-Trust-score-0412`. All experiments were conducted on a single NVIDIA A40 GPU (46GB VRAM, CUDA 12.5).

LLM USAGE

Large Language Models (LLMs) were used as a general-purpose writing assistant to translate text from the authors' mother tongue to English and to improve phrasing and clarity of the research paper.

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

## A    MATHEMATICAL FOUNDATIONS OF KERNEL DENSITY ESTIMATION

### A.1    FORMAL DEFINITION AND COMPONENTS

Given a sample of $n$ independent and identically distributed (i.i.d.) data points $X_1, \ldots, X_n$ from an unknown probability density function $f$, the kernel density estimator $\hat{f}(x; h)$ is defined as:

$$\hat{f}(x; h) = \frac{1}{n} \sum_{i=1}^{n} K_h(x - X_i) \tag{5}$$

where $K_h(z)$ is the scaled kernel function, defined as $K_h(z) = \frac{1}{h} K\left(\frac{z}{h}\right)$.

**Kernel Function $K(\cdot)$:** The kernel $K(u)$ is a non-negative real-valued function that generally satisfies $\int_{-\infty}^{\infty} K(u)\, du = 1$ and is symmetric around zero, i.e., $K(-u) = K(u)$. It determines the shape of the contribution of each data point to the global density estimation.

**Bandwidth Parameter $h$:** The bandwidth $h > 0$ is a crucial smoothing parameter that controls the spread of each kernel and, consequently, the smoothness of the resulting density estimate.

### A.2    KERNEL DENSITY ESTIMATION IN MULTIVARIATE DATA

The KDE framework naturally extends to multivariate data. For a d-dimensional sample $X_1, \ldots, X_n$, the multivariate KDE is given by:

$$\hat{f}_H(x) = \frac{1}{n} \sum_{i=1}^{n} |\mathbf{H}|^{-1/2} K(\mathbf{H}^{-1/2}(x - X_i)) \tag{6}$$

where $\mathbf{H}$ is the d×d bandwidth (or smoothing) matrix, which must be symmetric and positive definite.

**Bandwidth Matrix H:**   In the multivariate setting, $\mathbf{H}$ plays the role of a covariance matrix for the kernel. Critically, it controls not only the amount of smoothing but also its orientation, a fundamental distinction from univariate KDE.

**Distance Metric D:** The term $\mathbf{H}^{-1/2}(x - X_i)$ in Equation (6) implicitly defines a distance metric $\mathbf{D}$ between the point of interest $x$ and each data point $X_i$. Specifically:

$$D(x, X_i) = \left\| \mathbf{H}^{-1/2}(x - X_i) \right\|_2$$

where $\| \cdot \|_2$ is the Euclidean norm. This metric depends on the structure of $\mathbf{H}$:

- If $\mathbf{H} = h^2\mathbf{I}$ (isotropic scaling), $D(x, X_i)$ reduces to the Euclidean distance $\|x - X_i\|_2$.

- If $\mathbf{H}$ approximates the inverse covariance matrix $\boldsymbol{\Sigma}^{-1}$, $D(x, X_i)$ becomes the Mahalanobis distance, which accounts for correlations and scales between features.

A closer look at this formula reveals its reliance on distance. The term $(x - X_i)$ represents the vector difference between the point of interest $x$ and each data point $X_i$. The kernel function $K$ then assigns a weight to each data point based on this difference, with the highest weight given when the points are identical and the weight diminishing as they move farther apart. The shape of the kernel $K$ dictates how this distance is interpreted and how rapidly the influence of a data point fades with increasing separation.

This intrinsic link between KDE and distance allows us to use the density estimate directly as a confidence score. An example's confidence is directly related to its belonging within a given distribution; a high density value for a point $x$ implies that it is in close proximity to many other points (as defined by $\mathbf{D}$), making it a typical member of that distribution.

## B  EFFECT OF EMPIRICAL PRIORS ON KDE TRUST

In the main paper, we adopted the equal-prior simplification (see Sec. 3.3), which assumes that correct and incorrect predictions are equally likely within each class. This choice yields a confidence score that is independent of global model accuracy and easier to interpret as a geometry-driven measure of separability.

To analyse the impact of empirical priors, we first need to define global Polarisation as:

$$\text{Pol}_{\text{glob}} = \frac{1}{N} \sum_{i=1}^{N} \begin{cases} c_i & \text{if } y_i = 1, \\ 1 - c_i & \text{if } y_i = 0. \end{cases} \tag{7}$$

Unlike the Polarisation used in the main paper (Eq. 4), referred to here as balanced polarisation ($\text{Pol}_{\text{bal}}$) for simplicity, weights correct and incorrect predictions equally (regardless of class size). Whereas $\text{Pol}_{\text{glob}}$ assigns equal importance to each individual prediction; this makes it sensitive to the model's overall accuracy: a model with 90% correct predictions will have its global polarisation dominated by the 90% correct cases, even if the 10% incorrect cases are poorly separated. This distinction is needed when studying empirical priors, as $\text{Pol}_{\text{glob}}$ directly reflects the influence of the model's accuracy on confidence score geometry. In contrast, $\text{Pol}_{\text{bal}}$ ensures fairness across classes by normalising per-class contributions, making it ideal for evaluating geometry-driven separability in isolation. The deltas of the two polarisations scores are reported in Tables 3-6.

Table 3: KDE Trust metrics on MNIST for high vs low accuracy regimes, with/without Bayesian priors.

| Regime | Method | Pol. (bal.) | Pol. (glob.) | AUROC | Wasserstein |
|---|---|---|---|---|---|
| High (98%) | No Priors | $0.895 \pm 0.009$ | $0.870 \pm 0.004$ | $0.968 \pm 0.005$ | $0.789 \pm 0.018$ |
| | With Priors | $0.832 \pm 0.016$ | $0.960 \pm 0.002$ | $0.969 \pm 0.006$ | $0.664 \pm 0.032$ |
| $\Delta$ (With - No Priors) | – | $-0.063 \pm 0.019$ | $0.089 \pm 0.004$ | – | – |
| Low (76%) | No Priors | $0.895 \pm 0.015$ | $0.868 \pm 0.023$ | $0.978 \pm 0.006$ | $0.790 \pm 0.031$ |
| | With Priors | $0.907 \pm 0.014$ | $0.903 \pm 0.010$ | $0.978 \pm 0.006$ | $0.814 \pm 0.028$ |
| $\Delta$ (With - No Priors) | – | $0.012 \pm 0.021$ | $0.035 \pm 0.025$ | – | – |

Table 4: KDE Trust metrics on FashionMNIST for high vs low accuracy regimes, with/without Bayesian priors.

| Regime | Method | Pol. (bal.) | Pol. (glob.) | AUROC | Wasserstein |
|---|---|---|---|---|---|
| High (91%) | No Priors | $0.741 \pm 0.010$ | $0.864 \pm 0.017$ | $0.883 \pm 0.008$ | $0.482 \pm 0.021$ |
| | With Priors | $0.636 \pm 0.016$ | $0.893 \pm 0.014$ | $0.875 \pm 0.009$ | $0.272 \pm 0.032$ |
| $\Delta$ (With - No Priors) | – | $-0.105 \pm 0.019$ | $0.029 \pm 0.022$ | – | – |
| Low (69%) | No Priors | $0.773 \pm 0.028$ | $0.741 \pm 0.041$ | $0.899 \pm 0.022$ | $0.547 \pm 0.055$ |
| | With Priors | $0.800 \pm 0.022$ | $0.788 \pm 0.023$ | $0.914 \pm 0.017$ | $0.599 \pm 0.045$ |
| $\Delta$ (With - No Priors) | – | $0.026 \pm 0.036$ | $0.047 \pm 0.047$ | – | – |

Table 5: KDE Trust metrics on CIFAR-10 for high vs low accuracy regimes, with/without Bayesian priors.

| Regime | Method | Pol. (bal.) | Pol. (glob.) | AUROC | Wasserstein |
|---|---|---|---|---|---|
| High (89%) | No Priors | $0.780 \pm 0.006$ | $0.847 \pm 0.005$ | $0.894 \pm 0.005$ | $0.561 \pm 0.013$ |
| | With Priors | $0.683 \pm 0.013$ | $0.898 \pm 0.002$ | $0.894 \pm 0.005$ | $0.366 \pm 0.026$ |
| $\Delta$ (With - No Priors) | – | $-0.097 \pm 0.014$ | $0.051 \pm 0.005$ | – | – |
| Low (49%) | No Priors | $0.617 \pm 0.002$ | $0.621 \pm 0.003$ | $0.726 \pm 0.005$ | $0.234 \pm 0.004$ |
| | With Priors | $0.626 \pm 0.006$ | $0.632 \pm 0.006$ | $0.738 \pm 0.010$ | $0.252 \pm 0.013$ |
| $\Delta$ (With - No Priors) | – | $0.009 \pm 0.007$ | $0.010 \pm 0.006$ | – | – |

**Results.** We observe that including empirical priors systematically increases $\text{Pol}_{glob}$, since classes with high training accuracy dominate the weighted average. However, this comes at the cost of interpretability for high accuracy regimes, we find strong positive deltas for global polarisation but negative deltas for balanced polarisation. In other words, priors artificially inflate the polarisation score by encoding global accuracy, while reducing its ability to separate correct from incorrect predictions at the instance level. This makes the score less informative for human interpretation.

In contrast, on low accuracy regimes, the effect of priors is less pronounced. Both balanced and global Polarisation show positive deltas, indicating that here the prior information can help sharpen discrimination without degrading interpretability, provided the classifier's accuracy is sufficiently degraded. This pattern holds for most datasets, but IMDB presents a nuanced case: due to implementation constraints with the BERT architecture, our low-accuracy regime (82%) remains relatively high compared to other datasets (e.g., CIFAR-10 at 49%). Consequently, IMDB still exhibits a negative delta for balanced polarisation in this regime, though the magnitude is halved compared to the high-accuracy setting (92%). This attenuation aligns with our hypothesis: the impact of priors diminishes as accuracy decreases, but remains non-negligible when the low-accuracy regime is not sufficiently degraded.

Notably, the benefits of priors in low-accuracy regimes extend beyond polarisation. Both AUROC and Wasserstein distances also improve when priors are included in our KDE Trust score, except in high-accuracy models, where priors degrade performance. This suggests that the utility of empirical priors is context-dependent: they offer marginal benefits in genuinely low-accuracy settings but impair interpretability when the classifier is already highly accurate.

## C  Formulations of Evaluation Metrics

In this appendix, we detail the mathematical formulations of the metrics used to evaluate the performance of our confidence estimation method.

Table 6: KDE Trust metrics on IMDB for high vs low accuracy regimes, with/without Bayesian priors.

| Regime | Method | Pol. (bal.) | Pol. (glob.) | AUROC | Wasserstein |
|---|---|---|---|---|---|
| High (92%) | No Priors | $0.709 \pm 0.003$ | $0.696 \pm 0.008$ | $0.868 \pm 0.004$ | $0.419 \pm 0.006$ |
| | With Priors | $0.625 \pm 0.014$ | $0.866 \pm 0.003$ | $0.871 \pm 0.003$ | $0.250 \pm 0.027$ |
| $\Delta$ (With - No Priors) | – | $-0.085 \pm 0.014$ | $0.171 \pm 0.009$ | – | – |
| Low (82%) | No Priors | $0.664 \pm 0.036$ | $0.663 \pm 0.026$ | $0.804 \pm 0.038$ | $0.329 \pm 0.073$ |
| | With Priors | $0.620 \pm 0.082$ | $0.791 \pm 0.008$ | $0.812 \pm 0.044$ | $0.240 \pm 0.165$ |
| $\Delta$ (With - No Priors) | – | $-0.044 \pm 0.090$ | $0.128 \pm 0.028$ | – | – |

## C.1 AUROC (AREA UNDER THE RECEIVER OPERATING CHARACTERISTIC)

AUROC is a metric that measures a model's ability to discriminate between positive and negative predictions. It is calculated by plotting the ROC curve, which represents the True Positive Rate ($TPR$) as a function of the False Positive Rate ($FPR$) at different thresholds.

$$AUROC = \int_0^1 TPR(t) \cdot FPR'(t)\, dt \tag{8}$$

where $TPR$ and $FPR$ are defined as follows:

$$TPR = \frac{TP}{TP + FN}, FPR = \frac{FP}{FP + TN} \tag{9}$$

where $TP$ (True Positives), $FN$ (False Negatives), $FP$ (False Positives) and $TN$ (True Negatives).

## C.2 WASSERSTEIN DISTANCE (OR EARTH MOVER'S DISTANCE)

The first-order Wasserstein distance ($W_1$) between two probability distributions $P_r$ and $P_g$ is defined as the minimum amount of *work* required to transform one distribution into another. In our context, we compare the distribution of confidence scores for correct predictions, $P_{correct}$, and that for incorrect predictions, $P_{incorrect}$.

$$W(P_{correct}, P_{incorrect}) = \inf_{\gamma \in \Pi(P_{correct}, P_{incorrect})} \mathbb{E}_{(x,y) \sim \gamma}[\|x - y\|] \tag{10}$$

where $\Pi(P_{correct}, P_{incorrect})$ is the set of all joint distributions $\gamma(x, y)$ whose margins are $P_{correct}$ and $P_{incorrect}$, respectively. A small distance indicates that the distributions are close, i.e., that the correct and incorrect confidence scores are poorly separated.

## D HYPERPARAMETER ABLATION STUDY DETAILS

This appendix presents the complete numerical results of the hyperparameter ablation study conducted on all the datasets to determine the optimal configuration for our KDE-based confidence score.

The performance of each configuration was evaluated across multiple network layers (logits, $n - 1$, $n - 2$, and inputs). The study was conducted over the following hyperparameters:

- **Bandwidth**: Scott's rule, Silverman's rule, Grid Search (10 values log-spaced from 0.1 to 1)

- **Distance**: Euclidean, Manhattan, Mahalanobis

- **Kernel**: Gaussian, Exponential

## D.1 Summary of Findings and Final Configuration

To provide a clear overview, Table 7 summarises the optimal hyperparameter for each of our primary evaluation metrics.

Table 7: Optimal Hyperparameters for each individual metric across all studied datasets and accuracy regimes.

| Hyperparameter | AUROC | Polarisation | Wasserstein Distance |
|---|---|---|---|
| **Bandwidth (H)** | Scott/Silverman/Grid Search | Scott/Silverman | Scott/Silverman |
| **Distance (D)** | Mahalanobis | Mahalanobis | Mahalanobis |
| **Kernel (K)** | Gaussian | Gaussian | Gaussian |

Multiple configurations performed equivalently within statistical error margins. We selected **Silverman's rule**, **Gaussian kernel**, and **Mahalanobis distance**, as this configuration consistently ranked among the top-performing choices across metrics and seeds. It provided stable improvements in Wasserstein distance and Polarisation without degrading AUROC, and offered computational efficiency through a closed-form computation rather than an expensive grid search. While alternative configurations (e.g., Exponential kernel or Euclidean distance) achieved slightly higher AUROC in isolated cases, we adopt Silverman+Gaussian+Mahalanobis as our default since it yielded more robust performance across metrics and datasets.

A critical finding from our study is that the optimal hyperparameter configuration is remarkably consistent across all representation extraction layers. For a given evaluation metric, the best choice of $H$, $D$, and $K$ on the logits layer was also the best choice for the $n-1$, $n-2$, and input layers. This demonstrates the robustness of our approach and validates our final configuration for use in all experiments.

## D.2 Detailed Numerical Results

Tables 8–9 present a detailed ablation study on MNIST, isolating the impact of each hyperparameter while fixing the other two at (or near) their optimal values. For clarity, we focus on MNIST here; however, the same trends hold across all other datasets. Full results are available in our anonymous GitHub repository[1].

Table 8: Combined Ablation: Kernels, Bandwidths, Distances for MNIST, high accuracy.

| 98% Accuracy | Bandwidth | | | Kernel | | Distance | | |
|---|---|---|---|---|---|---|---|---|
| **Metrics and Layers** | Silverman | Scott | Grid Search | Gaussian | Exponential | Euclidean | Mahalanobis | Manhattan |
| **AUROC** | | | | | | | | |
| Logits | **0.968 ± 0.005** | **0.970 ± 0.004** | **0.970 ± 0.005** | **0.968 ± 0.005** | **0.972 ± 0.003** | 0.921 ± 0.005 | **0.968 ± 0.005** | 0.699 ± 0.007 |
| N-1 | 0.942 ± 0.004 | 0.945 ± 0.004 | 0.949 ± 0.003 | 0.942 ± 0.004 | 0.953 ± 0.003 | 0.753 ± 0.020 | 0.942 ± 0.004 | 0.670 ± 0.026 |
| N-2 | 0.930 ± 0.006 | 0.934 ± 0.006 | 0.941 ± 0.005 | 0.930 ± 0.006 | 0.944 ± 0.005 | 0.693 ± 0.020 | 0.930 ± 0.006 | 0.672 ± 0.017 |
| Inputs | 0.899 ± 0.004 | 0.904 ± 0.003 | 0.914 ± 0.002 | 0.899 ± 0.004 | 0.891 ± 0.004 | 0.828 ± 0.010 | 0.899 ± 0.004 | 0.658 ± 0.028 |
| **Polarisation** | | | | | | | | |
| Logits | **0.895 ± 0.008** | **0.894 ± 0.007** | **0.893 ± 0.008** | **0.895 ± 0.008** | 0.803 ± 0.007 | 0.772 ± 0.020 | **0.895 ± 0.008** | 0.648 ± 0.004 |
| N-1 | 0.845 ± 0.006 | 0.843 ± 0.006 | 0.852 ± 0.004 | 0.845 ± 0.006 | 0.757 ± 0.006 | 0.691 ± 0.019 | 0.845 ± 0.006 | 0.636 ± 0.023 |
| N-2 | 0.827 ± 0.010 | 0.825 ± 0.010 | 0.838 ± 0.008 | 0.827 ± 0.010 | 0.737 ± 0.007 | 0.647 ± 0.016 | 0.827 ± 0.010 | 0.650 ± 0.020 |
| Inputs | 0.781 ± 0.005 | 0.777 ± 0.005 | 0.792 ± 0.003 | 0.781 ± 0.005 | 0.664 ± 0.010 | 0.695 ± 0.004 | 0.781 ± 0.005 | 0.604 ± 0.012 |
| **Wasserstein Distance** | | | | | | | | |
| Logits | **0.789 ± 0.016** | **0.788 ± 0.015** | **0.786 ± 0.017** | **0.789 ± 0.016** | 0.605 ± 0.015 | 0.545 ± 0.040 | **0.789 ± 0.016** | 0.295 ± 0.009 |
| N-1 | 0.690 ± 0.013 | 0.685 ± 0.012 | 0.704 ± 0.008 | 0.690 ± 0.013 | 0.515 ± 0.013 | 0.381 ± 0.038 | 0.690 ± 0.013 | 0.272 ± 0.046 |
| N-2 | 0.654 ± 0.020 | 0.651 ± 0.021 | 0.675 ± 0.016 | 0.654 ± 0.020 | 0.474 ± 0.015 | 0.293 ± 0.031 | 0.654 ± 0.020 | 0.300 ± 0.039 |
| Inputs | 0.561 ± 0.010 | 0.553 ± 0.010 | 0.585 ± 0.006 | 0.561 ± 0.010 | 0.328 ± 0.020 | 0.390 ± 0.009 | 0.561 ± 0.010 | 0.209 ± 0.024 |

---

[1]`https://anonymous.4open.science/r/KDE-Trust-score-0412`

Table 9: Combined Ablation: Kernels, Bandwidths, Distances for MNIST, low accuracy.

| 76% Accuracy | Bandwidth | | | Kernel | | Distance | | |
|---|---|---|---|---|---|---|---|---|
| Metrics and Layers | Silverman | Scott | Grid Search | Gaussian | Exponential | Euclidean | Mahalanobis | Manhattan |
| **AUROC** | | | | | | | | |
| Logits | **0.978 ± 0.005** | **0.979 ± 0.005** | **0.977 ± 0.006** | **0.978 ± 0.005** | 0.975 ± 0.007 | 0.937 ± 0.014 | **0.978 ± 0.005** | 0.956 ± 0.015 |
| N-1 | **0.979 ± 0.005** | **0.980 ± 0.005** | **0.978 ± 0.006** | **0.979 ± 0.005** | 0.976 ± 0.006 | **0.975 ± 0.010** | **0.979 ± 0.005** | **0.914 ± 0.071** |
| N-2 | **0.979 ± 0.005** | **0.980 ± 0.005** | **0.978 ± 0.006** | **0.979 ± 0.005** | 0.976 ± 0.006 | 0.972 ± 0.011 | **0.979 ± 0.005** | 0.860 ± 0.087 |
| Inputs | **0.981 ± 0.003** | **0.982 ± 0.003** | **0.981 ± 0.004** | **0.981 ± 0.003** | 0.976 ± 0.005 | 0.964 ± 0.008 | **0.981 ± 0.003** | 0.749 ± 0.017 |
| **Polarisation** | | | | | | | | |
| Logits | **0.895 ± 0.014** | **0.892 ± 0.014** | **0.893 ± 0.014** | **0.895 ± 0.014** | 0.822 ± 0.021 | 0.700 ± 0.044 | **0.895 ± 0.014** | 0.824 ± 0.016 |
| N-1 | **0.893 ± 0.016** | **0.890 ± 0.016** | **0.892 ± 0.016** | **0.893 ± 0.016** | 0.820 ± 0.019 | **0.883 ± 0.010** | **0.893 ± 0.016** | **0.842 ± 0.079** |
| N-2 | **0.896 ± 0.016** | **0.893 ± 0.017** | **0.893 ± 0.017** | **0.896 ± 0.016** | 0.820 ± 0.019 | **0.897 ± 0.012** | **0.896 ± 0.016** | 0.786 ± 0.088 |
| Inputs | **0.903 ± 0.013** | **0.900 ± 0.013** | **0.903 ± 0.014** | **0.903 ± 0.013** | 0.817 ± 0.019 | **0.903 ± 0.015** | **0.903 ± 0.013** | 0.676 ± 0.013 |
| **Wasserstein Distance** | | | | | | | | |
| Logits | **0.790 ± 0.028** | **0.784 ± 0.029** | **0.785 ± 0.028** | **0.790 ± 0.028** | 0.644 ± 0.041 | 0.400 ± 0.088 | **0.790 ± 0.028** | 0.648 ± 0.032 |
| N-1 | **0.787 ± 0.031** | **0.780 ± 0.032** | **0.783 ± 0.032** | **0.787 ± 0.031** | 0.640 ± 0.039 | **0.767 ± 0.021** | **0.787 ± 0.031** | **0.684 ± 0.157** |
| N-2 | **0.791 ± 0.032** | **0.785 ± 0.033** | **0.786 ± 0.034** | **0.791 ± 0.032** | 0.641 ± 0.039 | **0.794 ± 0.024** | **0.791 ± 0.032** | 0.572 ± 0.176 |
| Inputs | **0.807 ± 0.025** | **0.799 ± 0.027** | **0.805 ± 0.028** | **0.807 ± 0.025** | 0.634 ± 0.038 | **0.805 ± 0.030** | **0.807 ± 0.025** | 0.353 ± 0.027 |

# E  Intrinsic Comparison of KDE Trust Across Different Layers

We provide a comprehensive visual representation of the KDE Trust score's performance under the selected hyperparameter configuration, across different internal representation of the DNN. Figure 5 shows that the logits layer generally yields the highest AUROC, followed by the $n-1$ and $n-2$ layers, with the input layer performing worst. These trends hold independently of model accuracy.

In low-accuracy regimes, however, an intriguing pattern emerges: performance improves across all layers for MNIST and FashionMNIST, with deeper layers showing more significant gains, ultimately matching the performance of the logits layer. Conversely, CIFAR-10 and IMDB exhibit degraded performance across layers, except for the input layer of CIFAR-10, which remains inferior yet shows relative improvement. We attribute the notable drop in all metrics for CIFAR-10 to the induced extremely low accuracy regime (49%), with higher accuracy leading to a lesser drop.

To mitigate the curse of dimensionality, we apply PCA with 10 components to high-dimensional layers, ensuring meaningful comparisons across layers. Our results demonstrate that the logits layer provides the best performance in the vast majority of cases across all datasets and accuracy regimes. Due to its robustness and simplicity (no PCA needed), we adopt it as our preferred choice for confidence estimation.

# F  Detail Results of KDE Trust across All Datasets and Accuracy Regimes

All numerical results of our experiments are reported in Tables 10-13.

Table 10: Confidence Score Evaluation on MNIST

| Accuracy | Method | AUROC | Polarisation | Wasserstein Distance |
|---|---|---|---|---|
| Low (76%) | Softmax Baseline | 0.808±0.038 | 0.573±0.027 | 0.147±0.054 |
| | Trust Score (Jiang et al., 2018) | 0.839±0.030 | 0.612±0.010* | 0.224±0.020* |
| | KDE Trust (ours) | **0.978±0.005** | **0.895±0.014** | **0.790±0.028** |
| High (98%) | Softmax Baseline | **0.969±0.002** | 0.651±0.005 | 0.301±0.009 |
| | Trust Score (Jiang et al., 2018) | 0.965±0.001 | 0.672±0.004* | 0.344±0.007* |
| | KDE Trust (ours) | **0.968±0.005** | **0.895±0.008** | **0.789±0.016** |

* Values winsorised to [0,1] based on 5th and 95th percentiles for fair comparison.

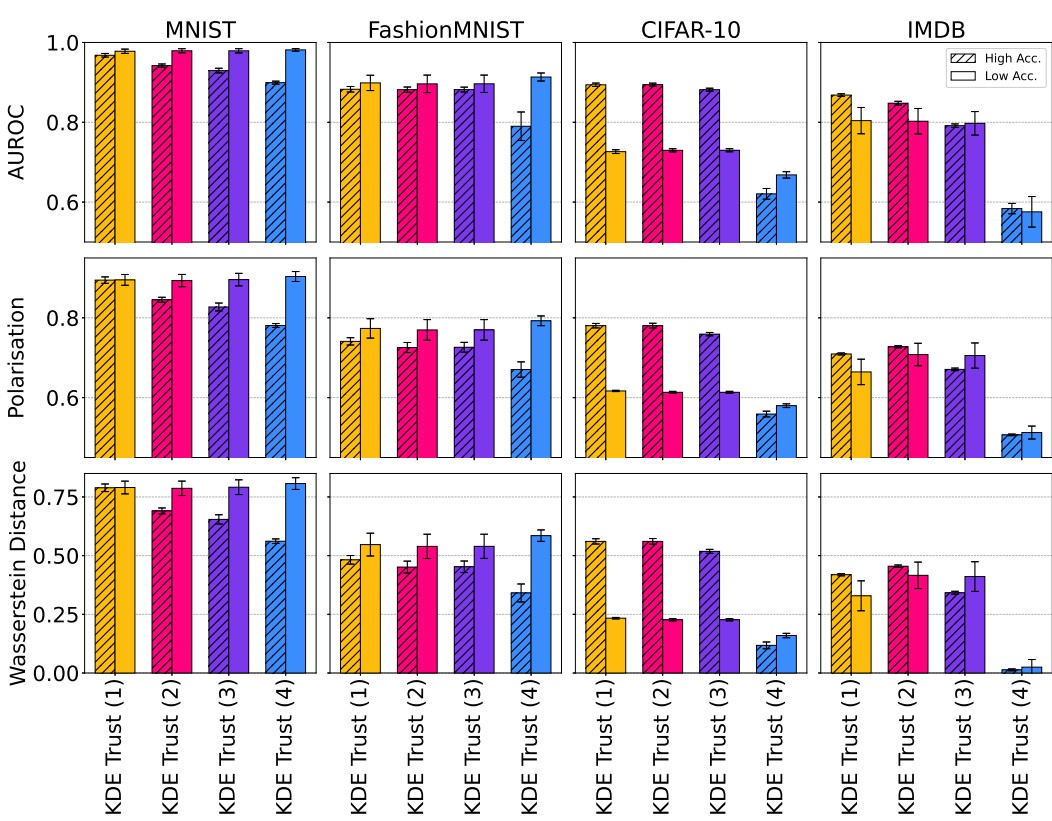

Figure 5: Intrinsic comparison of the proposed **KDE Trust** score across network layers, evaluated with AUROC, Polarisation, and Wasserstein distance between correct and incorrect predictions. **High accuracy regimes:** The Logits-based representation (1) and the penultimate layer (2) generally yield the strongest separation. **Low accuracy regimes:** For MNIST and FashionMNIST, performance improves across all layers, particularly in deeper layers, approaching logits layer performance. Conversely, CIFAR-10 and IMDB show a decrease in metrics across layers, except for the input layer of CIFAR-10, which remains inferior. All values are reported with 95% confidence intervals (CI). Legend: ● Logits (1) ● $n-1$ (2) ● $n-2$ (3) ● Input (4)

Table 11: Confidence Score Evaluation on FashionMNIST

| Accuracy | Method | AUROC | Polarisation | Wasserstein Distance |
|---|---|---|---|---|
| Low (69%) | Softmax Baseline | 0.796±0.015 | 0.607±0.010 | 0.214±0.019 |
| | Trust Score (Jiang et al., 2018) | 0.784±0.009 | 0.627±0.016* | 0.254±0.032* |
| | KDE Trust (ours) | **0.899±0.020** | **0.773±0.025** | **0.547±0.050** |
| High (91%) | Softmax Baseline | **0.909±0.002** | 0.612±0.006 | 0.225±0.012 |
| | Trust Score (Jiang et al., 2018) | 0.905±0.002 | 0.615±0.013* | 0.230±0.027* |
| | KDE Trust (ours) | 0.883±0.007 | **0.741±0.009** | **0.482±0.019** |

* Values winsorised to [0,1] based on $5^{th}$ and $95^{th}$ percentiles for fair comparison.

Table 12: Confidence Score Evaluation on CIFAR-10

| Accuracy | Method | AUROC | Polarisation | Wasserstein Distance |
|---|---|---|---|---|
| Low (49%) | Softmax Baseline | 0.699±0.010 | 0.572±0.004 | 0.145±0.008 |
| | Trust Score (Jiang et al., 2018) | 0.663±0.007 | 0.582±0.004* | 0.163±0.008* |
| | KDE Trust (ours) | **0.726±0.005** | **0.617±0.002** | **0.234±0.003** |
| High (89%) | Softmax Baseline | **0.904±0.006** | 0.615±0.004 | 0.230±0.008 |
| | Trust Score (Jiang et al., 2018) | **0.905±0.003** | 0.678±0.001* | 0.356±0.002* |
| | KDE Trust (ours) | 0.894±0.004 | **0.780±0.006** | **0.561±0.011** |

* Values winsorised to [0,1] based on $5^{th}$ and $95^{th}$ percentiles for fair comparison.

Table 13: Confidence Score Evaluation on IMDB

| Accuracy | Method | AUROC | Polarisation | Wasserstein Distance |
|---|---|---|---|---|
| Low (82%) | Softmax Baseline | 0.776±0.020 | 0.563±0.005 | 0.125±0.010 |
| | Trust Score (Jiang et al., 2018) | 0.731±0.048 | 0.597±0.013* | 0.194±0.027* |
| | KDE Trust (ours) | **0.804±0.034** | **0.664±0.033** | **0.329±0.065** |
| High (92%) | Softmax Baseline | 0.857±0.003 | 0.584±0.002 | 0.169±0.005 |
| | Trust Score (Jiang et al., 2018) | 0.834±0.005 | 0.635±0.003* | 0.270±0.005* |
| | KDE Trust (ours) | **0.868±0.003** | **0.709±0.003** | **0.419±0.005** |

* Values winsorised to [0,1] based on $5^{th}$ and $95^{th}$ percentiles for fair comparison.

