# OpenReview forum: "Signals of Trust: From Classification to Confidence"
_ICLR.cc/2026/Conference — Submitted to ICLR 2026_

### Official Review · Reviewer_s9oP · 2025-10-30

**Soundness:** 1
**Presentation:** 2
**Contribution:** 1
**Rating:** 2
**Confidence:** 5

**Summary:**

The authors propose a class-conditional distributional modelling of the layers of networks (primarily final logit layer) to infer confidence of correctness. They use KDE for modelling the class conditional distribution, and try PCA when dimensionality of the layers are higher. A more interpretable evaluation measure for confidence score is proposed called Polarisation,

**Strengths:**

The paper is decently written.

**Weaknesses:**

A significant issue is that basing the modelling on correct/incorrect would be catastrophic for scenarios where the model overfits on the training data, which is usually the case for most if not all well-trained models with decent complexity. For instance to make their MNIST experiments work I see in Table 2 the high and low accuracy are 76% and 98%. However, in practice, most networks very easily attain 100% training accuracy, yielding all correct examples. Either way, even if it was 99%, a huge class imbalance would also skew the accuracy of the KDE Estimator.

Most importantly fundamental baseline discussions are missing. The idea for density modelling for getting an overall confidence/uncertainty is absolutely not new and goes back decades. And there are so many papers in this space from before. One prominent example is "Deep k-Nearest Neighbors: Towards Confident, Interpretable and Robust Deep Learning" from 2018, which I see hasn't been referenced here. Furthermore, class-conditional density modelling using other approaches (in the context of pointwise mutual information P(x|y)/P(y)) has been proposed as well, also for the logit layers (same as this work) in a very recent work which focused on comparing different pointwise information measures ("Pointwise Information Measures as Confidence Estimators in Deep Neural Networks: A Comparative Study", ICML 2025).

Therefore, the comparisons are really sparse, only including trust score from 2018 and SoftMax. There is a very significant number of post-hoc approaches that have been proposed over the years. I'm afraid the datasets are quite limited as well.

**Questions:**

Please see weaknesses.

---

### Official Review · Reviewer_fBuS · 2025-10-31

**Soundness:** 2
**Presentation:** 2
**Contribution:** 1
**Rating:** 0
**Confidence:** 4

**Summary:**

The paper proposes KDE Trust, a post-hoc confidence scoring method that estimates P(correct|predicted=c) by fitting class-conditional KDEs over correctly and incorrectly classified training examples in representation space (typically logits). The method uses a Gaussian kernel with Silverman bandwidth and Mahalanobis distance to compute confidence as a likelihood ratio. The authors also introduce Polarisation, a metric measuring how close confidence scores are to 1 for correct predictions and 0 for incorrect ones.

**Strengths:**

- The approach is simple, post-hoc, and architecture-agnostic with clear intuition.
- Thorough ablations on kernel/bandwidth/distance choices and layer selection are provided.

**Weaknesses:**

The contribution is incremental and closely related to existing density/distance-based confidence methods (Trust Score, Mahalanobis confidence, energy scores). I am missing crucial baseline comparisons: Mahalanobis-based confidence, energy scores, Deep Ensembles, MC Dropout, ConfidNet/DDU/GRAM, and selective prediction with risk-coverage curves . Without these, it's difficult to assess the novelty and empirical merit of the approach.

The experimental setup is concerning. All experiments use small datasets (MNIST/FashionMNIST/CIFAR-10/IMDB) with simple architectures (small CNN/ResNet-18/DistilBERT). There are no results on CIFAR-100, ImageNet-scale models (ResNet-50/ViT), or modern vision backbones. More importantly, the "low-accuracy" regimes where KDE Trust shows gains are artificially constructed via early stopping, small batches, and inflated learning rates. This doesn't reflect how real systems fail and likely distorts the statistics of f_bad,c. In fact, on standard high-accuracy CIFAR-10, KDE Trust is below MSP/Trust Score in AUROC (0.894 vs 0.904/0.905) despite higher Polarisation. Can you evaluate on CIFAR-10-C/ImageNet-C, label noise, or natural distribution shifts rather than training-time degradation?

The Polarisation metric is problematic. It's easy to game and not invariant to monotone score transformations . The paper claims KDE Trust produces "posterior probability of correctness," yet the main variant uses equal priors. No reliability diagrams, ECE, or Brier/NLL are reported to support probabilistic interpretation.

The Trust Score baseline processing raises fairness concerns. You winsorize Trust Score to [0,1] at 5th-95th percentiles before computing Polarisation and Wasserstein distance. While AUROC is invariant to monotone transforms, distributional metrics are not - clipping baseline tails can materially inflate relative gains . Can you report Polarisation/Wasserstein for Trust Score without winsorization?

KDE scalability and robustness are inadequately addressed. For high-accuracy classes, f_bad,c may have very few samples, making KDE unstable.

The evaluation is missing standard diagnostics for instance-level confidence: AURC/risk-coverage curves for selective prediction, calibration metrics with reliability diagrams, FPR@95TPR for misclassification detection . These are needed to establish that KDE Trust improves decision quality under realistic conditions.

Can you clarify whether you intend the score to be probabilistic or ranking-based? If probabilistic, provide calibration analysis. If ranking-based, why introduce Polarisation instead of using established selective prediction metrics?

**Questions:**

see above

---

### Official Review · Reviewer_AFNz · 2025-11-01

**Soundness:** 2
**Presentation:** 2
**Contribution:** 2
**Rating:** 4
**Confidence:** 3

**Summary:**

The authors propose a KDE approach, which is a post-hoc method that models class-conditional distributions of correct and incorrect predictions in the deep neural network’s representation space, for better distinguishing the correct and incorrect predictions. A novel evaluation metric called Polarisation is also proposed in this context, which quantifies how close confidence scores are to 1 for correct classifications and 0 for incorrect ones.

**Strengths:**

1. The paper is structured well and easy to follow.

2. The approach is a light and post hoc method.

3. The presented experiments in the main body, including image and text classification cases, demonstrate the performance improvements of the proposed approach.

**Weaknesses:**

1. Given that this work presents a simple idea and a post hoc approach without rigorous theoretical analysis, it would be better to provide extensive empirical evidence to enhance the soundness of the method and convince the audience. Following these points, I believe that the current datasets and the backbones are quite simple; it is unclear how the method would perform in median or complex datasets  (e.g., the widely used baseline datasets CIFAR100 and ImageNet) and architectures.

2. I understand the motivation for having degraded-performance models to reflect the real-world settings, which often involve suboptimal model performance. However,  would settings such as *Early stopped at 1 epoch* be reasonable? As well as will a down to 49% model be considered in practice?

3. The author mentioned explicitly combining OOD detection with our score to enhance security as one of the future works. My concern is would this method would have the potential to do this by the proposed KDE (e.g., eq(3)), as OOD samples are far from the training distributions. I believe it is not difficult to demonstrate some verification results in this direction.

**Questions:**

Please refer to the weaknesses.

---

### Meta-Review · Area_Chair_NqoT · 2026-01-05

**Summary:**

This paper presents a method for performing post-hoc calibration of deep Neural networks to improve uncertainty quantification.  The method is based on kernel density estimation.  The authors also introduce a novel evaluation metric they call polarisation.

Unfortunately, the review scores were low (0, 2, 4) and the reviewers raised a variety of concerns with the work.  They were concerned that the polarisation metric was 'gamable' and not well justified. They found the experiments to not be compelling, missing standard baselines and on small benchmarks.  Finally, they were worried that the approach was incremental given many related approaches in the literature, and the authors didn't adequately discuss the relation to existing literature.

Unfortunately, the paper does not seem quite ready for publication.  Hopefully the reviews will be helpful in further developing the work.

**Reviewer Concerns:**

There was no rebuttal.

**Reviewer Scores:**

Unlikely any scores would have changed given the lack of rebuttal.

---

### Decision · Program_Chairs · 2026-01-26

Reject